# Association of sleep apnea with outcomes in peripheral artery disease: Insights from the PORTRAIT study

Qurat-ul-ain Jelani[1], Carlos Mena-Hurtado[1], Kensey Gosch[2], Moghniuddin Mohammed[2], Clementine Labrosciano[3], Christopher Regan[1], Lindsey E. Scierka[1], John A. Spertus[2,4], Sameer Nagpal[1], Kim G. Smolderen[1] *

1 Vascular Medicine Outcomes Program, Section of Cardiovascular Medicine, Department of Internal Medicine, Yale University School of Medicine, New Haven, Connecticut, United States of America, 2 Saint Luke's Mid America Heart Institute/University of Missouri – Kansas City, Kansas City, Missouri, United States of America, 3 Basil Hetzel Institute for Translational Research, Woodville, South Africa, 4 University of Missouri-Kansas City, Kansas City, Missouri, United States of America

* kim.smolderen@yale.edu

**Data Availability Statement:** Data contain potentially identifying information. The data are available upon request from the Yale Institutional Review Board. Institutional contact for data

## Abstract

### Background

Sleep apnea is a predictor of adverse cardiovascular outcome in many cardiovascular diseases but whether it is associated with worse health status outcomes or mortality in peripheral artery disease (PAD) is unknown.

### Methods

PORTRAIT is an international (US, Netherlands, Australia) prospective PAD registry that consecutively enrolled patients who presented with new-onset or recent exacerbations of PAD symptoms to any of 16 vascular specialty clinics. Health status was assessed upon presentation and at 12 months with the disease-specific Peripheral Artery Questionnaire (PAQ). Higher PAQ scores indicate better health status. A sequentially-adjusted hierarchical linear regression model examined the association between sleep apnea and 1-year PAQ symptoms, quality of life, and summary scores. Five-year survival curves by comorbid sleep apnea status for US patients were compared using the log-rank test.

### Results

The mean age of the 1204 PORTRAIT participants was 67.6 ± 9.4 years with 37.5% women and 8.3% (n = 100) having sleep apnea. Patients with sleep apnea were more likely to be from the US, more sedentary, and to have diabetes, obesity, coronary disease, more depressive symptoms and a history of prior peripheral interventions. Paradoxically, they also had higher ankle-brachial indices, but lower PAQ Summary scores at presentation and 12 months (41.2 ± 22.0 vs. 49. 9± 21.6 and 58.6 ± 27.9 vs. 71.3 ± 24.9, respectively, p = <0.05). The association between sleep apnea and 1-year health status persisted after

inquires: Gaelle Romain, PhD (Data Manager)
Email Address: gaelle.romain@yale.edu.

**Funding:** Dr. Smolderen is supported by an unrestricted research grant from Cardiva, Gore, Abbott, Merck, and Johnson & Johnson. She is a consultant for Optum Labs and Abbott. Dr. Spertus is a consultant to Bayer, Janssen, Merck, Myokardia, Novartis, Terumo and United Healthcare. He receives grant support from Janssen and Myokadia. He holds the copyright to the Peripheral Artery Questionnaire, Kansas City Cardiomyopathy Questionnaires, and the Seattle Angina Questionnaire. He serves on the Board of Blue Cross/Blue Shield of Kansas City. Dr. Mena is a consultant for Abbott Vascular, Boston Scientific, Cook, Medtronic, Cardinal Health, and Optum Labs. Other authors report no disclosure.

**Competing interests:** Dr. Smolderen is supported by an unrestricted research grant from Merck, Boston Scientific, Abbott Vascular and Terumo, and she is a consultant for Optum Labs. Dr. Spertus provides consultative services to Amgen, Merck, AstraZeneca, Novartis, Bayer, Janssen and United Healthcare. He has an equity interest in Health Outcomes Sciences and serves on the Board of Blue Cross Blue Shield of Kansas City. He owns the copyright to the Peripheral Artery Questionnaire, the Seattle Angina Questionnaire and the Kansas City Cardiomyopathy Questionnaire. Dr. Mena-Hurtado is a consultant for Bard, Cook, Medtronic, Abbott, Boston Scientific and Cardinal Health. Other authors report no disclosure.

multivariable adjustment, but there were no differences in all-cause mortality over 5 years (28.0% vs. 23.4%, p = 0.76).

## Conclusion

In patients presenting with PAD, comorbid sleep apnea is independently associated with worse health status over time. Future studies should test whether better treatment of sleep apnea can improve the health status of patients with PAD.

## Clinical trial registration

NCT01419080

## Introduction

Peripheral artery disease (PAD) is characterized by atherosclerosis of the aorta, iliac and lower extremity arteries and affects around 8.5 million peoples in the United States (US) [1] and ~236 million people worldwide [2]. It is associated with significant mortality, morbidity and health care costs [3–5]. Typical atherosclerotic risk factors underlie PAD including diabetes mellitus, smoking, hypertension, obesity, and hyperlipidemia [6–10]. A less known comorbidity that may occur in patients with PAD, is sleep apnea [11]. Sleep apnea has been studied more extensively in coronary artery and heart failure patients wherein it has been linked with higher cardiovascular morbidity and mortality [12–18], adverse quality of life outcomes [19], stress [20], anxiety and depression [21–23]. In contrast, sleep apnea is under-studied in PAD with understanding of its relationship with key outcomes such as health status, psychological factors, and mortality. Measurement of health status outcomes, especially disease-specific outcomes, is of clinical relevance in that it directly quantifies a patient's perspectives about their symptoms, functioning and quality of life as related to a disease or its treatment [24].

Given this current incomplete understanding, we aimed to study the association between sleep apnea and PAD outcomes, including 1-year PAD-specific health status, psychological outcomes (depressive symptoms and anxiety) and 5-year mortality outcomes. We hypothesize that patients with comorbid sleep apnea and PAD will have worse outcomes as compared with their counterparts who do not present with sleep apnea. Understanding this potentially novel and treatable risk factor in PAD, would allow us to better risk-stratify and support patients at risk of experiencing worse PAD outcomes.

## Methods

### Participants and study design

Patients included in this study were enrolled from the PORTRAIT (Patient-centered Outcomes Related to Treatment Practices in Peripheral Artery Disease Investigating Trajectories) registry, which is an international, prospective, observational study designed to address gaps in our knowledge about care of patients with PAD and their outcomes [25]. A total of 1275 patients presenting with a new diagnosis or exacerbation of PAD symptoms were enrolled in this registry from June 2011 to December 2015. Patients were enrolled from 16 vascular specialty clinics, with ten from the United States, five from the Netherlands and one from Australia.

Patients were enrolled if they met the following inclusion criteria: (1) Doppler resting ankle-brachial index (ABI) $\leq$ 0.90 or a significant drop in post-exercise ankle pressure of $\geq$ 20 mmHg [26, 27]; (2) patients aged $\geq$18 years; (3) new-onset or recent exacerbation of exertional leg symptoms, regardless of whether symptoms were typical or atypical (buttock, thigh, hip or calf pain, numbness or discomfort inhibiting the patient's ability to walk distances). Symptoms were categorized as typical or atypical as described in medical charts by the treating physician. Patients with a non-compressible ABI $\geq$ 1.30, those who underwent a lower-limb revascularization procedure in the past year (angioplasty, bypass surgery, atherectomy, endarterectomy) for the ipsilateral leg relative to where the patient was currently having symptoms, patients with critical limb ischemia (Rutherford Classification 4–6) [28], patients who could speak neither English or Spanish or Dutch, and patients with hearing impairment or current imprisonment were excluded.

This study was conducted in accordance with the Declaration of Helsinki. For the POR-TRAIT Registry, the study was approved by the Institutional Review Board (IRB) of the coordinating site St. Luke's Mid America Heart Institute and subsequently each participating site provided study approval. All patients provided informed consent before enrollment in the study. Consent was obtained from study participants via written or verbal phone consent. The consent process was witnessed and documented. The Strengthening the Reporting of Observational Studies in Epidemiology guidelines for reporting observational studies were used [29]. For this specific study, IRB approval was granted by the Yale Institutional Review Board and additional consent from the subjects already enrolled in the PORTRAIT Study was not required. Information on mortality was collected for patients from the USA only through the National Death Index.

De-identified data is available for data sharing via a request to the PORTRAIT Principal Investigator, Dr. Kim Smolderen per study protocol. All requests for data are reviewed by the PORTRAIT Publications Committee, a team of sub-investigators involved in the original PORTRAIT registry.

## Data collection and study definitions

Baseline characteristics including demographics, socioeconomic status, psychosocial characteristics, and health status were obtained via interview at the initial visit. Patients' symptoms, medical history, comorbidities and PAD diagnostic information were abstracted from medical records. Information on presence or absence of any sleep apnea (regardless of the etiology) and treatment with continuous positive airway pressure (CPAP) ("Yes" or "No") was obtained from medical records. No screening questionnaire for sleep apnea or formal tests such as nocturnal polysomnography was conducted to screen for or confirm the diagnosis of sleep apnea in these patients either at enrollment or during follow-up.

Follow-up health status, depressive symptoms and anxiety assessments were conducted at 3, 6 and 12 months by a centralized call-center.

## Assessment of outcomes

Disease-specific health status outcomes were measured by the Peripheral Artery Questionnaire (PAQ). The PAQ is a 20-item, validated, PAD-specific, multi-dimensional health status instrument that measures six health status domains relevant to patients with PAD: physical function, symptoms, symptom stability, social limitations, treatment satisfaction, and quality of life [30]. One item identifies the most symptomatic leg and the remaining 19 items are answered using ordinal response scales. A summary score is calculated as the average of the physical limitation,

symptoms, quality of life, and social functioning scores. Scores range from 0 to 100 points, with higher scores indicating better functioning.

Generic health status was assessed by the EQ-5D and consists of a descriptive section (EQ-5D Index score) and a visual analog scale (EQ-5D VAS score). For our study, the EQ-5D VAS score was used to measure patients' overall health status [31]. The EQ-5D VAS is a 20-cm Visual Analog Scale that ranges from the worst (a score of 0) to the best (a score of 100) imaginable state, with higher scores indicating better health status.

All-cause mortality information for US patients was derived from querying the National Death Index [32]. Mortality was determined for US patients only as records were not available for the Dutch and Australian cohorts. The National Death Index provides vital status of patients from the US [33]wherein causes of death are listed by International Classification of Disease (ICD) 10 codes [32]. Censoring of mortality information was up to 12/31/2018.

Depressive symptoms were assessed through the Patient Health Questionnaire (PHQ-8) [34], an eight-item depression screening scale that has designed based on the Diagnostic and Statistical Manual IV criteria for a major depressive disorder. A PHQ-8 score $\geq$ 10 has 88% sensitivity and specificity to detect major depression.

Generalized anxiety symptoms were measured using the Generalized Anxiety Disorder Scale (GAD-2) [35], a short screening tool that consists of the first two questions of the GAD-7 scale, which has good reliability, factorial and structural validity to screen for generalized anxiety disorder [36]. In the GAD-2, participants are asked about how often they feel nervous and how often they are not able to stop worrying. A GAD score of > 3 has a 86% sensitivity and 83%% specificity to detect a possible generalized anxiety disorder [36].

Other variables were collected including socioeconomic status as determined by questions regarding highest level of education (less than high school, high school, college/post-graduate), avoidance of care due to cost (yes vs. no), and financial resources left at the end of the month (some, just enough, not enough). Information on medical history, risk factors, and comorbidities consisted of history of dyslipidemia, stroke, hypertension, coronary artery disease, myocardial infarction, percutaneous coronary interventions, coronary artery bypass grafting, body mass index, congestive heart failure, chronic kidney disease, chronic lung disease, diabetes, and smoking. PAD disease severity was assessed by using the resting ABI and extent of claudication (mild, moderate, severe) at baseline. We also collected information patients' symptom presentation (new symptoms versus exacerbation of symptoms), duration of pain and lesion characteristics (lesion location, lesion site and laterality of symptomatic leg). History of lower-extremity amputations and history or surgical or endovascular lower-extremity interventions were also collected. Treatment strategy at 3 months following patients' enrollment was abstracted from patients' medical records and categorized as non-invasive (medical therapy only) or invasive (percutaneous or surgical intervention).

## Statistical analysis

We compared baseline characteristics between patients with or without sleep apnea using Chi-square or Fisher's Exact tests for categorical variables and Student's t-tests for continuous variables. Categorical variables were presented as frequencies and percentages and continuous variables are shown as means with standard deviations or medians with interquartile ranges.

Unadjusted mean health status scores and psychological assessments (PAQ summary score, PAQ symptoms score and PAQ physical functioning, EQ5D-VAS, PHQ-8 and GAD-2 scores) were summarized by comorbid sleep apnea status at baseline, 3, 6, and 12 months and compared through ANOVA tests.

The association between comorbid sleep apnea status and disease-specific health status outcomes (PAQ summary score, PAQ symptoms score and PAQ physical functioning) over the course of 1-year by comorbid sleep apnea status was modeled through deriving pooled estimates of the mean difference (aggregating 3-, 6-, and 12-month health status information) from hierarchical linear regression models. Models were sequentially adjusted, the first step included the unadjusted effect, with a random effect for site, step 2, additionally adjusted for patient demographics (age, sex, race and country); step 3, additionally adjusted for socioeconomic variables (education status, insurance, work for pay and avoiding care due to cost), in step 4, PAD disease severity/characteristics at presentation were added (resting ABI value, functional limitation, duration of symptoms, new onset and unilateral disease), step 5 added cardiovascular risk factors/comorbidities (obesity, hypertension, coronary artery disease, congestive heart failure, history of musculoskeletal problems (back pain), and chronic lung disease), step 6 added psychological variables (PHQ-8 scores), and finally, adjustments for baseline health status were made. Variables included in the models have previously been shown to be predictors of poor health status outcomes in patients with PAD [29]. We also tested the interaction between sleep apnea and two other variables: (1) CPAP use (yes versus no) and (2) initial management strategy (revascularization versus conservative management without revascularization) to the unadjusted models to verify whether differences in health status outcomes existed for those who were treated with CPAP or revascularization versus those who were not.

To examine the association between sleep apnea and 5-year all-cause mortality, we constructed Kaplan-Meier curves to compare the age-adjusted risk of all-cause mortality over the following 5-years, in patients with and without sleep apnea, and tested the survival curves using the log-rank test.

All statistical analyses were performed using SAS 9.4 (SAS Institute Inc, Cary, NC). All statistical tests were two-tailed and significance was determined using an $\alpha$-level of 0.05.

### Missing data

Complete covariate data was available on 82% of patients, with 17% missing one covariate and 1% missing two covariates. The covariates with the highest number of missingness were "duration of pain" (n = 175, 15%) and "PHQ-8 depression score" (n = 29, 2%). Data were assumed to be missing at random and were imputed using a single imputation model that contained all variables used in the multivariable model.

### Results

The final cohort consisted of 1204 patients, of which 8.3% (n = 100) had comorbid sleep apnea (Table 1). The mean age of the entire cohort was 67.7 ± 9.4 years with 37.5% women and 82.1% whites. Patients with sleep apnea were predominantly from the US (in part because of highest enrollment) and were more likely to have education at high school level or above. There were no differences between the two groups in terms of socioeconomic status including marital status, insurance, avoiding care due to cost or difficulty getting care.

Overall, patients with PAD and sleep apnea had a higher burden of cardiovascular risk factors and comorbidities, including dyslipidemia, hypertension, diabetes, coronary artery disease, prior myocardial infarction, prior percutaneous coronary intervention, and chronic heart failure (all p = < 0.05). Furthermore, patients with sleep apnea were more likely to be obese, to have chronic back pain and were more likely to be sedentary. A total of 53 (53.0%) patients with sleep apnea were being treated with CPAP at the time of enrollment in the study. Higher proportions of patients with vs. without sleep apnea had a history of peripheral vascular

**Table 1. Baseline characteristics including co-morbidities, disease severity at presentation and treatment allocation after enrollment, stratified by presence or absence of sleep apnea.**

| | Sleep Apnea | | Total | P-value |
|---|---|---|---|---|
| | **Yes**<br>**N = 100 (8.3%)** | **No**<br>**N = 1104 (91.7%)** | **N = 1204 (100%)** | |
| *Socio-Demographics/Socioeconomic Status* | | | | |
| Age | | | | |
| Mean ± SD | 68.5 ± 8.8 | 67.5 ± 9.4 | 67.6 ± 9.4 | 0.32 |
| Sex | | | | |
| Male | 68 (68.0) | 685 (62.0) | 753(62.5) | 0.24 |
| Race: White/Caucasian | 78 (78.0) | 911 (82.5) | 989 (82.1) | 0.26 |
| Country | | | | |
| USA | 83 (83.0) | 665 (60.2) | 748 (62.1) | |
| Netherlands | 7 (7.0) | 359 (32.5) | 366 (30.4) | |
| Australia | 10 (10.0) | 80 (7.2) | 90 (7.5) | <0.001 |
| Married | 57 (58.8) | 651 (59.1) | 708 (59.1) | 0.94 |
| Education High School or above | 83 (83.8) | 743 (67.9) | 826 (69.2) | < 0.001 |
| Work for pay | 20 (20.0) | 269 (24.5) | 289 (24.1) | 0.32 |
| Insurance: Have insurance, Medicare or Medicaid | 99 (99.0) | 1096 (99.3) | 1195 (99.3) | 0.54 |
| Avoid care due to cost | 15 (15.2) | 155 (14.1) | 170 (14.2) | 0.78 |
| BMI | | | | |
| Mean ± SD | 34.6 ± 7.5 | 28.4 ± 5.6 | 29.0 ± 6.1 | <0.001 |
| Activity During Leisure Time | | | | |
| Sedentary | 51 (51.5) | 433 (40.1) | 484 (41.0) | |
| Mild | 31 (31.3) | 360 (33.3) | 391 (33.1) | |
| Moderate/Strenuous | 17 (17.2) | 288 (26.6) | 305 (25.8) | 0.045 |
| *Cardiac Risk Factors and Comorbidities* | | | | |
| CAD | 60 (60.0) | 473 (42.8) | 533 (44.3) | < 0.001 |
| Dyslipidemia | 95 (95.0) | 863 (78.2) | 958 (79.6) | < 0.001 |
| Hypertension | 92 (92.0) | 876 (79.3) | 968 (80.4) | 0.002 |
| Diabetes | 49 (49.0) | 348 (31.5) | 397 (33.0) | < 0.001 |
| Prior Angina | 17 (17.0) | 151 (13.7) | 168 (14.0) | 0.36 |
| Prior MI | 27 (27.0) | 201 (18.2) | 228 (18.9) | 0.031 |
| Prior PCI | 42 (42.0) | 227 (20.6) | 269 (22.3) | < 0.001 |
| Prior CABG | 24 (24.0) | 211 (19.1) | 235 (19.5) | 0.24 |
| Smoke status | | | | |
| Never | 11 (11.0) | 118 (10.7) | 129 (10.7) | |
| Former | 62 (62.0) | 565 (51.3) | 627 (52.2) | |
| Current | 27 (27.0) | 419 (38.0) | 446 (37.1) | 0.08 |
| Chronic kidney disease | 16 (16.0) | 117 (10.6) | 133 (11.0) | 0.09 |
| Prior CVA | 11 (11.0) | 83 (7.5) | 94 (7.8) | 0.21 |
| Prior TIA | 7 (7.0) | 57 (5.2) | 64 (5.3) | 0.43 |
| Chronic lung disease | 19 (19.0) | 189 (17.1) | 208 (17.3) | 0.63 |
| Congestive heart failure | 21 (21.0) | 103 (9.3) | 124 (10.3) | < 0.001 |
| Chronic back pain | 21 (21.0) | 144 (13.0) | 165 (13.7) | 0.03 |
| Cancer | 12 (12.0) | 109 (9.9) | 121 (10.0) | 0.49 |
| *PAD Disease History* | | | | |
| Non-healing ulcer | 1 (1.0) | 15 (1.4) | 16 (1.3) | 1.00 |
| Amputation | 1 (1.0) | 13 (1.2) | 14 (1.2) | 1.00 |
| PAD bypass | 7 (7.0) | 89 (8.1) | 96 (8.0) | 0.71 |
| PAD endarterectomy | 4 (4.0) | 32 (2.9) | 36 (3.0) | 0.53 |

*(Continued)*

**Table 1.** (Continued)

| | Sleep Apnea | | Total | P-value |
|---|---|---|---|---|
| | **Yes**<br>**N = 100 (8.3%)** | **No**<br>**N = 1104 (91.7%)** | **N = 1204 (100%)** | |
| PAD atherectomy | 8 (8.0) | 20 (1.8) | 28 (2.3) | 0.001 |
| PAD angioplasty | 34 (34.0) | 202 (18.3) | 236 (19.6) | < 0.001 |
| Peripheral vascular intervention | 39 (39.0) | 288 (26.1) | 327 (27.2) | 0.005 |
| Cilostazol | 5 (5.0) | 69 (6.3) | 74 (6.2) | 0.61 |
| Antiplatelet | 78 (78.0) | 742 (67.6) | 820 (68.4) | 0.031 |
| Statin | 77 (77.0) | 758 (69.0) | 835 (69.7) | 0.09 |
| ACEI/ARB | 74 (74.0) | 640 (58.0) | 714 (59.3) | 0.001 |
| *PAD Characteristics at Presentation* | | | | |
| ABI | | | | |
| Mean ± SD | 0.75 ± 0.19 | 0.66 ± 0.18 | 0.67 ± 0.19 | <0.001 |
| Duration of Pain | | | | |
| < 1 Month | 0 (0.0) | 27 (2.9) | 27 (2.6) | |
| 1–6 Months | 21 (24.4) | 286 (30.3) | 307 (29.8) | |
| 7–12 Months | 11 (12.8) | 171 (18.1) | 182 (17.7) | |
| >12 Months | 54 (62.8) | 459 (48.7) | 513 (49.9) | 0.05 |
| Symptoms | | | | |
| New-onset | 33 (33.0) | 601 (54.4) | 634 (52.7) | |
| Exacerbation | 67 (67.0) | 503 (45.6) | 570 (47.3) | < 0.001 |
| Symptom presentation | | | | |
| Typical | 78 (81.3) | 877 (86.5) | 955 (86.0) | |
| Atypical | 18 (18.8) | 137 (13.5) | 155 (14.0) | 0.16 |
| Lesion Site | | | | |
| Proximal | 28 (28.0) | 308 (28.1) | 336 (28.1) | |
| Distal | 33 (33.0) | 318 (29.0) | 351 (29.4) | |
| Both | 39 (39.0) | 469 (42.8) | 508 (42.5) | 0.67 |
| Symptomatic Leg: Both | 60 (60.0) | 556 (50.4) | 616 (51.2) | 0.06 |
| Rutherford category | | | | |
| Mild claudication (Rutherford Class 1) | 20 (20.2) | 250 (22.9) | 270 (22.7) | |
| Moderate claudication (Rutherford Class 2) | 54 (54.5) | 533 (48.9) | 587 (49.3) | |
| Severe claudication (Rutherford Class 3) | 25 (25.3) | 308 (28.2) | 333 (28.0) | 0.55 |
| *PAD Treatment After Enrollment* | | | | |
| Cilostazol | 12 (12.0) | 129 (11.7) | 141 (11.8) | 0.94 |
| Statin | 81 (81.0) | 885 (80.6) | 966 (80.6) | 0.92 |
| ACEI/ARB | 76 (76.0) | 648 (58.7) | 724 (60.1) | <0.001 |
| Antiplatelet Treatment | 90 (90.0) | 956 (86.6) | 1046 (86.9) | 0.40 |
| Smoking Cessation Counseling | 20 (20.0) | 309 (28.0) | 329 (27.3) | 0.40 |
| Supervised Exercise Referral | 12 (12.0) | 251(22.7) | 263(21.8) | 0.013 |
| Invasive treatment (3- Months) | 15 (15.6) | 213 (20.4) | 228 (20.0) | 0.26 |
| Weight Management Counseling | 6(6.0) | 46 (4.2) | 52 (4.3) | 0.43 |
| PHQ-8 Score ≥10 | 24 (24.7) | 162 (15.0) | 186 (15.8) | 0.01 |
| GAD-2 Score | 17 (17.2) | 167(15.2) | 184 (15.4) | 0.61 |

All values are described as "N (%)" unless described otherwise.

Continuous variables compared using Student's T-test.

Categorical variables compared using chi-square or Fisher's exact test.

Abbreviations: SD: Standard Deviation; USA: United States of America; BMI: Body Mass Index; PAD: Peripheral Artery Disease; CAD: Coronary Artery Disease; MI: Myocardial Infarction; PCI: Percutaneous Coronary Intervention; CABG: Coronary Artery Bypass Grafting; TIA: Transient Ischemic Attack; CVA: Cerebrovascular Accident; PHQ-8: Patient Health Questionnaire; GAD: Generalized Anxiety Disorder; EQ-5D: European Quality of Life Questionnaire.

interventions and were on anti-platelet and Angiotensin Converting Enzyme Inhibitors (ACE-I) and Angiotensin Receptor blockers (ARB). Although patients presented with higher ABIs at presentation, they were more likely to present with exacerbation of symptoms (Table 1). Patients with sleep apnea were also less likely to be referred for supervised exercise treatment (SET) as compared with those who did not have sleep apnea (Table 1).

### Health status outcomes

Unadjusted mean scores for both PAD-specific and generic health status outcomes were consistently lower upon presentation and throughout the 1-year follow-up. For the PAD-specific scales, the mean difference ranged from 7.4 to 12.7, and the mean difference for the EQ5D scores ranged from 5.2 to 6.8 (Table 2).

### Psychological outcomes

Both at baseline and at 3, 6, and 12 months follow-up, patients with sleep apnea had significantly higher mean depressive symptom scores on the PHQ-8, reflective of a higher depressive symptom burden. This increased burden was also reflected in the dichotomized PHQ-8 $\geq$ 10 rates (24.0% vs 15.0%, p = < 0.05). There were no differences in anxiety levels by comorbid sleep apnea status (Tables 1 and 2).

### PAD-specific health status models

After one year, patients with sleep apnea versus those without had worse health status outcomes as determined by PAQ summary scores despite sequential adjustment for demographics, socioeconomic status, PAD disease severity, comorbidities, psychosocial factors and baseline health status. The adjusted aggregate mean difference associated with comorbid sleep apnea status and PAD-specific health status outcomes following is depicted in Fig 1. Patients with sleep apnea versus those without, had on average ~ 10-point difference on the unadjusted PAQ summary score scales. The effect size in the PAQ summary score remained robust following sequential adjustments and attenuated to ~ 5 points following addition of the psychological and baseline health status information. In the fully adjusted models, the association between sleep apnea and worse health status persisted. The association between sleep apnea and PAQ domains including symptoms and quality of life did not remain significant after sequential adjustment. There were no differences in health status outcomes between sleep apnea patients using CPAP versus those with sleep apnea without CPAP treatment or between sleep apnea patients treated with revascularization versus those with sleep apnea who were managed conservatively (without revascularization) in the unadjusted model so no further modeling including these interaction terms was performed.

### 5-year mortality

Long-term all-cause mortality rate for the overall US cohort was 24%. Median follow-up time was 4.1 years (Interquartile Range 3.5–4.7 years). Patients with PAD who had comorbid sleep apnea versus those without had similar mortality rates (28.0% versus 23.4%, log-rank test, p = 0.76) (Fig 2). For our sample size, we have 80% power to detect a 12.6% difference in mortality between those with versus those without sleep apnea.

### Discussion

The association between comorbid sleep apnea in PAD and subsequent outcomes has never been evaluated. For the first time, we demonstrate that patients with PAD that present with

**Table 2. Psychosocial factors and unadjusted PAQ symptom, quality of life and summary score at baseline, 3, 6 and 12 months after enrollment in PORTRAIT registry.**

| | Sleep Apnea N (%) | | Total | P-value |
| | Yes<br>N = 100 (8.3%) | No<br>N = 1104 (91.7%) | N = 1204 (100%) | |
|---|---|---|---|---|
| *Psychosocial Factors* | | | | |
| **PHQ-8 Depression Score** | | | | |
| Baseline | 6.3 ± 5.3 | 4.6 ±5.0 | 4.7 ± 5.0 | 0.001 |
| 3 Months | 4.9 ±5.3 | 3.5 ±4.4 | 3.6 ±4.5 | 0.003 |
| 6 Months | 5.0 ± 5.4 | 3.2 ± 4.2 | 3.4 ± 4.3 | <0.001 |
| 12 Months | 4.9 ±4.6 | 3.1 ±4.1 | 3.3 ± 4.1 | <0.001 |
| **GAD Anxiety Score** | | | | |
| Baseline | 1.2 ± 1.6 | 1.0 ±1.6 | 1.0 ± 1.6 | 0.25 |
| 6 Months | 0.8 ± 1.7 | 0.7 ± 1.4 | 0.7 ± 1.4 | 0.59 |
| 12 Months | 0.8 ± 1.5 | 0.6 ± 1.3 | 0.6 ± 1.3 | 0.19 |
| *Health Status Outcomes* | | | | |
| **PAQ Symptoms** | | | | |
| Baseline | 40.0 ±23.0 | 44.3 ± 22.8 | 43.9 ± 22.8 | 0.07 |
| 3 Months | 53.7± 29.5 | 58.2 ± 28.5 | 57.8 ± 28.6 | 0.13 |
| 6 Months | 54.8 ±28.9 | 62.0 ± 29.1 | 61.4 ± 29.1 | 0.02 |
| 12 Months | 51.0 ± 30.1 | 63.0 ± 29.7 | 62.0 ± 29.9 | <0.001 |
| **PAQ Quality of Life** | | | | |
| Baseline | 41.9 ± 26.7 | 51.3 ± 25.7 | 50.5 ± 25.9 | <0.001 |
| 3 Months | 58.4 ±30.0 | 67.9 ± 27.3 | 67.1 ± 27.6 | 0.001 |
| 6 Months | 64.1 ± 27.6 | 71.7 ± 27.0 | 71.1 ± 27.1 | 0.01 |
| 12 Months | 61.1 ± 30.0 | 72.5 ± 27.2 | 71.6 ± 27.6 | <0.001 |
| **PAQ Summary Score** | | | | |
| Baseline | 41.2 ±22.0 | 49.9 ± 21.6 | 49.2 ± 21.7 | <0.001 |
| 3 Months | 59.7 ±27.2 | 67.3 ± 24.4 | 66.7 ± 24.7 | 0.003 |
| 6 Months | 62.9 ± 23.1 | 70.3 ± 24.5 | 69.7 ± 24.4 | 0.005 |
| 12 Months | 58.6 ±27.9 | 71.3 ± 24.9 | 70.2 ± 25.4 | < 0.001 |
| **EQ-5D Scores** | | | | |
| Baseline | 59.9 ± 21.7 | 66.7 ± 19.0 | 66.1 ± 19.4 | <0.001 |
| 3 Months | 64.8 ±19.1 | 70.0 ± 18.6 | 69.6 ± 18.7 | 0.008 |
| 6 Months | 65.5 ± 18.9 | 70.7 ± 18.0 | 70.2 ± 18.1 | 0.008 |
| 12 Months | 65.2 ±20.2 | 70.7 ± 17.4 | 70.2 ± 17.7 | 0.005 |

All values are presented as "Mean ± standard deviation" unless otherwise specified.

Continuous variables compared using Student's T-test.

Categorical variables compared using chi-square or Fisher's exact test.

Abbreviations: PHQ-8: 8-Item Patient health Questionnaire; GAD: Generalized Anxiety Disorder; PAQ: Peripheral Artery Questionnaire; EQ-5D: European Quality of Life Questionnaire.

sleep apnea have a uniquely increased burden of cardiovascular comorbidities and risk profile. Their health status scores lag behind their counterparts who do not present with sleep apnea, and they are more vulnerable to experience depressive symptoms. While long-term mortality rates were comparable, patients with PAD and comorbid sleep apnea do not achieve similar PAD specific health status over the course of 1 year following their PAD work-up, thereby representing a barrier to attaining optimal health status recovery following PAD treatment.

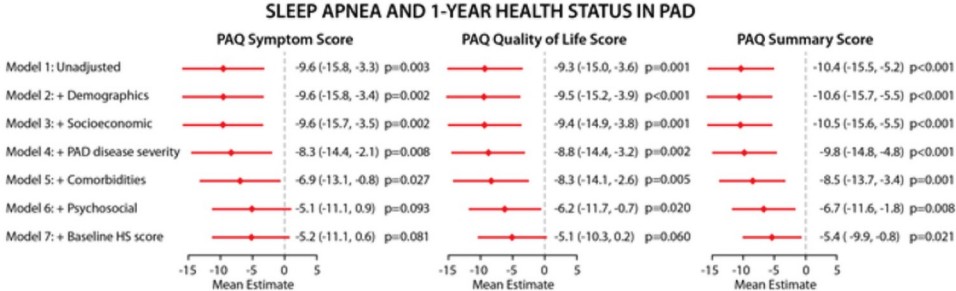

**Fig 1. Twelve month mean differences for the association between sleep apnea and PAQ health status scores (PAQ symptoms, quality of life, and summary scores).** The models were sequentially adjusted for (1) patient demographics (age, sex, race and country), (2) socioeconomic variables (education status, insurance, work for pay and avoiding care due to cost), (3) PAD disease severity characteristics (ABI value, functional limitation, duration of symptoms and unilateral disease), (4) cardiovascular risk factors/comorbidities (obesity, hypertension, coronary artery disease, chronic heart failure, musculoskeletal variables (back pain and osteoarthritis) and chronic lung disease, (5) psychological variables (PHQ-8 scores), and finally, (6) adjustment for baseline health status. Abbreviations: PAQ: Peripheral Artery Questionnaire; PAD: Peripheral Artery Disease; CI: Confidence Interval.

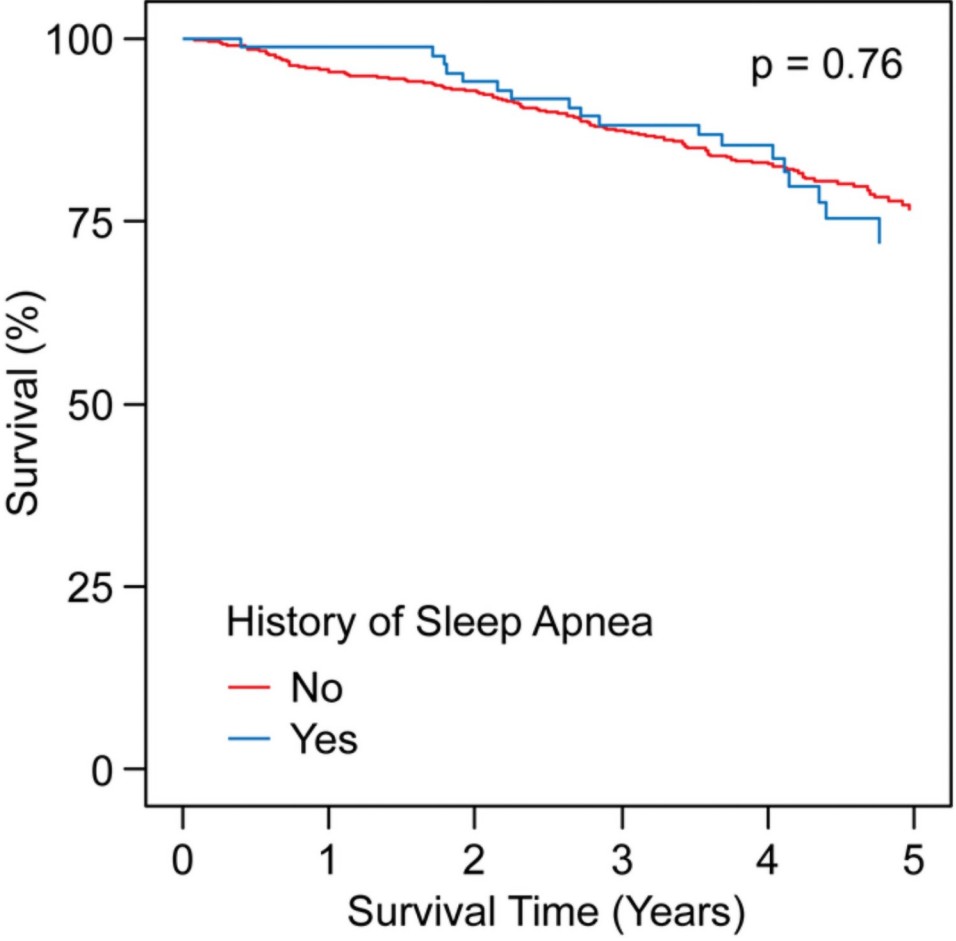

**Fig 2. Kaplan Meier 5-year survival curves by sleep apnea status.** The curves were compared by the log-rank test.

Obstructive sleep apnea (OSA) is the most common form of sleep apnea in cardiovascular disease patients and also in the overall population [37–39]. Sleep apnea and its link with outcomes has been well-studied in other forms of cardiovascular disease including in those with atherosclerotic risk factors such as hypertension, chronic heart failure, atrial fibrillation, stroke, and coronary artery disease [40–43] Despite the increasing prevalence and health implications of both PAD and sleep apnea both nationally and globally [1, 44, 45], the association between sleep apnea and key PAD outcomes have not been studied. The unique PORTRAIT cohort includes patients with a recent PAD diagnosis, who are tracked longitudinally for both patient-centered outcomes as well as long-term prognosis and allowed us to investigate these associations.

Patients with PAD and comorbid sleep apnea present with a clustering of cardiovascular risk factors that makes them a particularly high-risk subgroup as compared with patients who do not present with comorbid sleep apnea. These findings are consistent with findings from coronary patient profiles [46–50]. According to some studies, the manifestation of other cardiovascular risk factors increases progressively with sleep apnea severity [50–52], and is prognostic of adverse coronary and cerebrovascular outcomes [18, 53, 54].

While our study was not able to replicate the association between comorbid sleep apnea and prognostic outcomes [13–15, 53–55] such as mortality because of power limitations, the clustering of cardiovascular risk factors in the PAD population that presents with sleep apnea deserves attention. Regardless of whether sleep apnea can be considered an independent risk factor of adverse outcomes for PAD or whether it is a marker of underlying atherosclerotic risk, it is important to emphasize that treatment of sleep apnea (weight loss, behavioral therapy including alcohol cessation, CPAP and surgical correction) [56] can potentially lead to improved outcomes, including patients' health status and psychological outcomes such as depressive symptoms in patient with hypertension, atrial fibrillation and heart failure [40–42]. In addition, CPAP treatment has also been shown to improve endothelial function after initiation [57].

The notion that treatment of sleep apnea can improve outcomes such as health status and quality of life is important [43, 58–64]. We found that patients with PAD who presented with comorbid sleep apnea had a significantly blunted health status, symptom, and quality of life recovery trajectory over year following their PAD diagnosis, even after adjusting for depressive symptoms, disease severity, and comorbidities. At the same time, despite higher BMIs in the sleep apnea cohort, only 6% of the sleep apnea cohort was referred to a weight loss management program, and patients with sleep apnea overall, were less likely referred to a structured exercise program, leaving many areas of improvement in the care of these patients as ways to help maximize outcomes following a PAD diagnosis.

Findings of diminished health status are consistent with coronary and non-coronary populations who are dealing with sleep apnea [43, 58, 65, 66]. These impairments could be attributed to subjective sleepiness [65], poor sleep quality [67], and depression [68]. Along with a higher burden of depressive symptoms, another physiological explanation for worse disease-specific outcomes in our patients maybe the enhanced pain sensitivity seen in patients with sleep disruptions [58, 67, 68] and chronic recurrent hypoxia [58]. We also observed increased vulnerability in terms of patients' psychological outcomes, and with that, identify another area that warrants further attention in patients with sleep apnea. Patients with sleep apnea are known to have a higher prevalence of depression (20–40%) [22, 69] compared with the general population (3–15%) [70]. While there is evidence for improvement in depressive symptoms after direct treatment of sleep apnea [43, 71], it is important to offer further evaluation and treatment for depression in patients with PAD, as depression is an important risk factor for adverse cardiovascular and health status outcomes in and of its own [72–75].

The complex behavioral reality of managing multiple clustered chronic comorbidities like sleep apnea, clustering of atherosclerotic risk factors as well as psychosocial risk factors underscore the need for multidisciplinary chronic disease management programs for PAD, with an emphasis on managing life-style factors, weight reduction, and structured exercise programs. In PAD specifically, these needs seem to be particularly unmet and expertise need to be further built to be able to successfully manage the clustering of diseases and provide patients with adequate support.

Our study must be interpreted in the light of several limitations. Although sleep apnea is commonly undiagnosed, we did not prospectively perform formal diagnostic testing or screening questionnaires for sleep apnea, relying on information collected from medical charts, thus exposing our study to the risk of misclassification bias. Our study did not collect information on the severity of sleep apnea, and thus we could not evaluate the association between sleep apnea severity and outcomes. Similarly, we were unable to account for the type of sleep apnea (obstructive versus central sleep apnea), although prior work shows us that the majority of cases are explained by obstructive sleep apnea, especially in atherosclerotic populations [37]. We were unable to assess whether the duration of having comorbid sleep apnea was associated with the severity of PAD, as increased duration of disease has been linked with increasing intima thickness in carotid stenosis [76]. In addition, our study is observational in nature and while we were able to adjust for a wide range of relevant patient characteristics, the possibility of residual confounding remains. Lastly, our study enrolled patients from a select set of vascular specialty clinics and thus our findings may not necessarily extend to other settings of care where patients with PAD are being seen.

In conclusion, our study shows that sleep apnea patients with PAD are much sicker with a higher burden of cardiovascular risk factors and comorbidities. Patients with PAD and comorbid sleep apnea present also with a much more vulnerable mental health profile as reflected in the depressive symptom burden. Importantly, an independent association between sleep apnea and worse PAD-specific health status profiles was noted, warranting an increased awareness for sleep apnea as an expression of clustering of cardiovascular risk profiles and thereby representing a subgroup that is particularly at risk of adverse PAD outcomes.

## Supporting information

**S1 File.**
(DOCX)

## Author Contributions

**Conceptualization:** Qurat-ul-ain Jelani, Carlos Mena-Hurtado, Kim G. Smolderen.

**Data curation:** Kensey Gosch.

**Formal analysis:** Kensey Gosch.

**Funding acquisition:** Kim G. Smolderen.

**Investigation:** Carlos Mena-Hurtado, Kim G. Smolderen.

**Methodology:** Qurat-ul-ain Jelani, Carlos Mena-Hurtado, Kensey Gosch, Kim G. Smolderen.

**Project administration:** Qurat-ul-ain Jelani, Lindsey E. Scierka.

**Resources:** Kim G. Smolderen.

**Software:** Kensey Gosch.

**Supervision:** Carlos Mena-Hurtado, John A. Spertus, Kim G. Smolderen.

**Validation:** Kensey Gosch.

**Visualization:** Qurat-ul-ain Jelani, Kensey Gosch.

**Writing – original draft:** Qurat-ul-ain Jelani, Carlos Mena-Hurtado, Lindsey E. Scierka, Kim G. Smolderen.

**Writing – review & editing:** Qurat-ul-ain Jelani, Carlos Mena-Hurtado, Kensey Gosch, Moghniuddin Mohammed, Clementine Labrosciano, Christopher Regan, Lindsey E. Scierka, John A. Spertus, Sameer Nagpal.

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
