## [Decision Letter · Decision Letter 0]

17 Jun 2021

PONE-D-21-11479

Association of Sleep Apnea with Outcomes in Peripheral Artery Disease: Insights from the PORTRAIT Study

PLOS ONE

Dear Dr. Scierka,

Thank you for submitting your manuscript to PLOS ONE. After careful consideration, we feel that it has merit but does not fully meet PLOS ONE’s publication criteria as it currently stands. Therefore, we invite you to submit a revised version of the manuscript that addresses the points raised during the review process.

We look forward to receiving your revised manuscript.

Kind regards,

Carmine Pizzi

Academic Editor

PLOS ONE

1. Please ensure that your manuscript meets PLOS ONE's style requirements, including those for file naming. The PLOS ONE style templates can be found athttps://journals.plos.org/plosone/s/file?id=wjVg/PLOSOne_formatting_sample_main_body.pdf and https://journals.plos.org/plosone/s/file?id=ba62/PLOSOne_formatting_sample_title_authors_affiliations.pdf

2. Thank you for including your ethics statement:  "This study was conducted in accordance with the Declaration of Helsinki. The Institutional Review Board of each participating site provided study approval and all patients provided informed consent before enrollment in the study. The Strengthening the Reporting of Observational Studies in Epidemiology guidelines for reporting observational studies were used.".   

a.) Please amend your current ethics statement to include the full name of the ethics committee/institutional review board(s) that approved your specific study.

b.) Please provide additional details regarding participant consent. In the ethics statement in the Methods and online submission information, please ensure that you have specified what type you obtained (for instance, written or verbal, and if verbal, how it was documented and witnessed). If your study included minors, state whether you obtained consent from parents or guardians. If the need for consent was waived by the ethics committee, please include this information.

'Dr. Smolderen is supported by an unrestricted research grant from Merck, Boston Scientific, Abbott Vascular and Terumo, and she is a consultant for Optum Labs.

Dr. Spertus provides consultative services to Amgen, Merck, AstraZeneca, Novartis, Bayer, Janssen and United Healthcare. He has an equity interest in Health Outcomes Sciences and serves on the Board of Blue Cross Blue Shield of Kansas City. He owns the copyright to the Peripheral Artery Questionnaire, the Seattle Angina Questionnaire and the Kansas City Cardiomyopathy Questionnaire.

Dr. Mena-Hurtado is a consultant for Bard, Cook, Medtronic, Abbott, Boston Scientific and Cardinal Health.

Other authors report no disclosure.' 

Reviewer #1: To:

Title: “Association of Sleep Apnea with Outcomes in Peripheral Artery Disease: Insights from the PORTRAIT Study”

- The use of questionnaires can be considered as a limitation of the study. Please discuss this point into a dedicated limitation section.

- The authors can also consider the impact of CPAP therapy on PAD outcomes. This would be really interesting as Ciccone et al. [Int J Cardiol. 2012 Jul 26;158(3):383-6] previously demonstrated the reversibility of endothelial function after CPAP therapy.

- The duration of sleep apnea may impact on the severity of the PAD. Please discuss such a point in relation to the paper from Ciccone MM et al. Respir Med. 2012 May;106(5):740-6.

Reviewer #2: This is an interesting and original study; however, as often happens in studies analyzing big data, the results should be addressed in a more specific way. I have some questions for the authors:

1. Methods page 5 line 18: please provide the reference about Rutherford Classification

2*. Could you please provide a comparison between patients with different Rutherford classes (3,4,5 and 6) with and without sleep apnea? PAD includes very different patterns of disease which can not be put together in the same analysis.

3*. The authors should also stratify the results between patients who were revascularized and those who were not in order to make the results more homogeneous.

---

## [Author Response · Author response to Decision Letter 0]

29 Jul 2021

Responses to Reviewer #1 – 

1. The use of questionnaires can be considered as a limitation of the study. Please discuss this point into a dedicated limitation section.

We respectfully disagree with the reviewer on this comment. As research and clinical management of chronic disease populations shifts away from focusing only biomedical factors, methods have been developed to capture more patient-centered information. Validated patient reported outcomes measures (including the PAQ, EQ5D, GAD, PHQ etc used in this study) are high quality and rigorously tested tools developed using the classic test theory and also adopted by standards set by the FDA for the measurements of patient reported outcomes. These validated instruments are the gold standard for capturing patient reported outcomes and are a reliable and valid way to collect and study disease-specific health status and psychological constructs assessed in this study. 

2. The authors can also consider the impact of CPAP therapy on PAD outcomes. This would be really interesting as Ciccone et al. [Int J Cardiol. 2012 Jul 26;158(3):383-6] previously demonstrated the reversibility of endothelial function after CPAP therapy. 

We agree that adding the implications of treatment with CPAP to the discussion would add to the manuscript. We have added the following commend and citation to the Discussion section [page 16, line 274]: 

“Regardless of whether sleep apnea can be considered an independent risk factor of adverse outcomes for PAD or whether it is a marker of underlying atherosclerotic risk, it is important to emphasize that treatment of sleep apnea (weight loss, behavioral therapy including alcohol cessation, CPAP and surgical correction) [56] can potentially lead to improved outcomes, including patients’ health status and psychological outcomes such as depressive symptoms in patient with hypertension, atrial fibrillation and heart failure [40-42]. In addition, CPAP treatment has also been shown to improve endothelial function after initiation [57].”

3. The duration of sleep apnea may impact on the severity of the PAD. Please discuss such a point in relation to the paper from Ciccone MM et al. Respir Med. 2012 May;106(5):740-6.

We agree that the duration of sleep apnea may have an impact on the severity of PAD. We have added this as a limitation in the Discussion section [page 17, line 313] as our study is unable to associate the duration of sleep apnea with features of the patient’s PAD as follows: 

“We were unable to assess whether the duration of having comorbid sleep apnea was associated with the severity of PAD, as increased duration of disease has been linked with increasing intima thickness in carotid stenosis [76].”

Responses to Reviewer #2 – 

1. Methods page 5 line 18: please provide the reference about Rutherford Classification.

The citation for the Rutherford Classification (Rutherford, 1997) has been added.

2. “Could you please provide a comparison between patients with different Rutherford classes (3,4,5 and 6) with and without sleep apnea? PAD includes very different patterns of disease which can not be put together in the same analysis.” 

We agree that there is variation in PAD by Rutherford Classification, however we are unable to provide the analysis requested as the PORTRAIT Registry only included patients with symptomatic claudication (Rutherford Class 1, 2, and 3), and Rutherford 4-6 was an exclusion criterion. In our results, we have included a comparison between those with and without sleep apnea by symptom severity (mild, moderate, and severe claudication symptoms), and we found this association to not be statistically significant (p = 0.55). The Rutherford Classification labels corresponding to those symptoms (Class 1, 2, and 3 respectively) have been added to Table 1 for clarity [page 9-10 with changes on page 10 and 11]. 

3. “The authors should also stratify the results between patients who were revascularized and those who were not in order to make the results more homogeneous.” 

We stratified the health status outcomes by management strategy (revascularization versus conservative management without revascularization) and did not find a statistically significant difference. This information was added to the manuscript as follows: 

Methods [page 8, line 166]

“We also tested the interaction between sleep apnea and two other variables: (1) CPAP use (yes versus no) and (2) initial management strategy (revascularization versus conservative management without revascularization) to the unadjusted models to verify whether differences in health status outcomes existed for those who were treated with CPAP or revascularization versus those who were not.”

Results [page 13, line 226]

“There were no differences in health status outcomes between sleep apnea patients using CPAP versus those with sleep apnea without CPAP treatment or between sleep apnea patients treated with revascularization versus those with sleep apnea who were managed conservatively (without revascularization) in the unadjusted model so no further modeling including these interaction terms was performed.”

---

## [Decision Letter · Decision Letter 1]

19 Aug 2021

Association of sleep apnea with outcomes in peripheral artery disease: Insights from the PORTRAIT study

PONE-D-21-11479R1

Dear Dr. Smolderen,

We’re pleased to inform you that your manuscript has been judged scientifically suitable for publication and will be formally accepted for publication once it meets all outstanding technical requirements.

Kind regards,

Carmine Pizzi

Academic Editor

PLOS ONE

---

## [Editor Report · Acceptance letter]

31 Aug 2021

PONE-D-21-11479R1 

Association of sleep apnea with outcomes in peripheral artery disease: Insights from the PORTRAIT study 

Dear Dr. Smolderen:

I'm pleased to inform you that your manuscript has been deemed suitable for publication in PLOS ONE. Congratulations! Your manuscript is now with our production department. 

Kind regards, 

on behalf of

Prof Carmine Pizzi 

Academic Editor

PLOS ONE